# High Q GaN/SiC-Based SAW Resonators for Humidity Sensor Applications

**DOI:** 10.3390/mi16020150

**Published:** 2025-01-28

**Authors:** Dan Vasilache, Claudia Nastase, George Boldeiu, Monica Nedelcu, Catalin Parvulescu, Adrian Dinescu, Alexandru Muller

**Affiliations:** National Institute for Research and Development in Microtechnologies, 077190 Bucharest, Romania; dan.vasilache@imt.ro (D.V.); george.boldeiu@imt.ro (G.B.); catalin.parvulescu@imt.ro (C.P.); adrian.dinescu@im.ro (A.D.); alexandru.muller@imt.ro (A.M.)

**Keywords:** SAW sensor, humidity, SiO_2_

## Abstract

This paper presents the simulation and experimental results for high-frequency surface acoustic wave (SAW) sensors for humidity detection. The SAW structures with a wavelength of 680 nm are fabricated on GaN/SiC and presented two resonance frequencies: ~6.66 GHz for the Rayleigh propagation mode and ~8 GHz for the Sezawa mode. A SiO_2_ thin layer (~50 nm thick) was employed for the functionalization of the SAW. Relative humidity characterization was performed in the range of 20–90%. The SAW sensors achieved high values of humidity sensitivity for both adsorption and desorption. The Sezawa mode showed about 2.5 times higher humidity sensitivity than the Rayleigh mode: 17.2 KHz/%RH versus 6.17 KHz/%RH for adsorption and 8.88 KHz/%RH versus 3.79 KHz/%RH for desorption.

## 1. Introduction

The use of electronic devices is becoming essential in everyday life, both on a personal level (household appliances, mobile phones, etc.), and on a professional level (industrial equipment, health, etc.). Along with the increase in performance and complexity of the devices, there has also been an increase in their sensitivity in relation to external factors, one of the most important being humidity (the amount of moisture in the air).

High humidity levels can lead to moisture seeping into sensitive electronic components, causing corrosion and short circuiting, particularly damaging circuit boards, connectors, and other exposed parts. Similarly, low humidity levels can create problems by building up static electricity, leading to electrostatic discharges that can cause immediate damage or latent failure, significantly reducing the lifespan of electronic devices. Maintaining optimal humidity levels can help mitigate these risks. For these reasons, the precise measurement and control of humidity levels is crucial in various industrial, medical, and environmental applications.

Humidity sensors are transducers that convert the amount of water vapor (H_2_O) into a measurable parameter. The literature shows that humidity sensors operating at different temperatures have been reported, envisaging robotic applications and water condensation prevention [1,2,3,4]. Various sensing modalities have been developed in the last decades, which detect relative humidity (RH) in terms of resistance, capacitance, or refractive index (RI). A high-performance humidity sensor should have high sensitivity, long-term durability, fast response, and low cost and operate over a wide range of humidity levels and temperatures [5].

Different types of sensors have been developed, which can be classified as electrical (capacitive, resistive, etc.), mechanical (hygrometers and resonance), optical (through transmission, reflection, or quenching), or integrated (FET, LC or RC circuits and integrated mechanical sensors with electronics) [5].

Surface acoustic wave (SAW) sensors are sensitive to external perturbations that affect their intrinsic (viscosity, mass density, stiffness, electrical conductivity, and permittivity) and extrinsic (temperature and pressure) parameters. Surface acoustic waves are acoustic modes confined to a thin near-surface region of the propagation medium. The acoustic energy confinement makes these waves extremely sensitive to surface perturbations. Surface acoustic waves have been intensively used in sensor manufacturing due to their compatibility with wireless data transmission and battery-less operation, major advantages for devices performing in harsh environmental conditions. Wireless data systems for SAW-type sensors have been developed in [6,7] and adapted for humidity sensors in [8,9,10,11,12]. In addition, the SAW-based sensor also features small size, high sensitivity, good stability, high resolution, good repeatability, and good radiation resistance [13,14].

The use of SAW devices as sensors has been demonstrated in numerous fields, with or without the use of a functionalization layer; SAWs have been used as temperature [15] and pressure [16] sensors and for the detection of humidity [17] and different types of gas [18,19].

Most of the SAW-based sensors have used resonators manufactured on classical bulk non-semiconductor piezoelectric substrates, such as quartz, langasite, or lithium niobate. These materials have excellent piezoelectric properties, but the quality of the surface makes it very difficult to develop advanced nanolithographic technologies. This limits the resonance frequencies bellow 2.5 GHz. An increase in the SAW resonance frequency is important not only for communication applications of SAW devices but also in sensor applications, as an increased resonance frequency enhances the sensitivity of the SAW sensor. A solution to increase the resonance frequency is to use III-Nitride piezoelectric semiconductor layered structures to fabricate the SAW devices [20,21,22,23,24,25]. GaN/Si, GaN/SiC, and GaN/Sapphire [26] are layered structures fully compatible with nanolithography and the micromachining process that can be introduced into the fabrication protocol of acoustic devices. Also, integration with other circuit components into GaN-based microwave monolithic integrated circuits is possible. In the last years, these piezoelectric materials have become attractive for high-sensitivity temperature [27] and pressure sensors [28] based on acoustic devices. The considered layered structures are “slow on fast” structures (the phase velocity in the piezoelectric overlayer is smaller than in the substrate), and a superior higher resonance frequency mode (Sezawa) appears. This represents the advantage for most applications (including humidity sensing applications), as higher resonance frequency operation increases sensitivity. For the experiments presented in this work, we chose GaN/SiC, as it has two advantages compared to GaN/Si structures: (i) The GaN layer deposited on SiC is of higher quality compared to silicon due to fewer problems regarding the lattice mismatch which appears during the MOCVD growing process, and (ii) there is the potential possibility to use the sensor at very high temperatures.

The operation of humidity detection is realized by a sensing film overlaid on the surface of the SAW device. Any perturbation on the surface due to the adsorption/desorption of water molecules affects the intrinsic parameters (mass density, thickness, conductivity, and elasticity) of the sensing film, which alters the wave characteristics [17].

When the SAW devices are used as gas or humidity sensors, the measured responses arise from perturbations in wave propagation characteristics. Consequently, the interactions between the surface waves and the functionalization layer deposited over the SAW, containing the adsorped gas (or vapor), determine the velocity change in the response [29]. SAW–film interactions arise from mechanical coupling due to a mass loading mechanism, caused by the translation of surface mass by the SAW surface displacement. The main physical mechanism of SAW interaction with humidity is the mass loading mechanism, having as an effect the change in phase velocity.

The fractional frequency changes (∆f/f_0_) are equal to the fractional velocity changes, and the equation for the mass change in SAW propagation velocity is ∆f/f_0_ = ∆v/v_0_ = −c_m_f_0_ ∆m/A, where c_m_ is the mass sensitivity factor, a constant specific to the piezoelectric layer, and ∆m/A is the mass change per surface unit [30]. The sensitivity calculated for an SAW device used as a gas sensor is S = d∆f/d (∆m/A) = −c_m_f_0_^2^. The sensitivity of mass loading-based devices is directly proportional to the square of its operating frequency for gas and humidity detection SAW sensors [31]. For humidity sensors, the sensing performance is quantified as absolute sensitivity (S_a_ = Δf/ΔRH) and relative sensitivity (S_r_ = S_a_/f_0_), where ΔRH is the change in relative humidity (RH) [32].

Different types of functionalization layers have been used to fabricate SAW humidity sensors, such as polymers [33], 2D layers (like GO—Graphene Oxide) [34], or oxides (Co_3_O_4_, CuO, ZnO, TiO_2_, and SiO_2_) [35,36,37]. However, published results seem to show a better stability of structures over time using different types of oxides compared to polymers or 2D materials.

It is the aim of this work to evaluate high-frequency SAW resonators on GaN/SiC (resonance frequency higher than 6 GHz) with a high-quality factor (Q) for humidity sensing applications [31]. The Q factor shows the level of losses that occur through different mechanisms [38], affecting the performance of the SAW devices [39]. The current work proposes SAW resonators with a 680 nm wavelength (170 nm electrode width), and two different numbers of electrode pairs (75 and 150) are analyzed in relation to the quality factor improvement [40].

A thin SiO_2_ layer used to functionalize the humidity sensor has the role of adsorbing moisture from the air, which leads to a change in mass and consequently to a resonant frequency shift. The porosity of the SiO_2_ layer and its thickness are factors that substantially influence the sensor’s response [35]. The SiO_2_ layer also insulates the IDT’s structure of metal electrodes, preventing them from short circuiting. The advantages presented are related to the reliability or repeatability of the measurements, and a major advantage is the compatibility with the silicon device fabrication technology.

## 2. SAW Humidity Sensor—Theoretical Background

Simulating this type of sensor is challenging, with the main problem appearing on the coupling of the equations; in addition to the piezoelectric effect equations, we also had Frick’s diffusion equation:(1)$$\frac{\partial c}{\partial t}+\nabla \left(-D\nabla c\right)=0$$(2)$$D=D_{0}exp\left(-\frac{U}{k_{B}T}\right)$$
where *c* is the concentration, *D* is the diffusion coefficient, *U* is the activation energy, and *k_B_* is the Boltzmann’s constant.

Another problem is the lack of certain material parameters. One of the simulation methods refers to changing the density of the sensitive material and the thickness of it, using the relationships below [41,42]:(3)$$\rho \left(C_{v}\right)=\frac{\rho_{0}+k\cdot C_{v}}{1+k\cdot C_{v}/\rho_{V}}$$(4)$$h\left(C_{v}\right)=h_{0}\left(1+\frac{k\cdot C_{v}}{\rho_{v}}\right)$$
where $\rho_{0}$ and $h_{0}$ are the initial density and initial height of the material, $C_{v}$ is the vapor concentration in the air, $\rho_{v}$ is the vapor density, and *k* is the air/coating film coefficient.

The simulation was performed in COMSOL MultiPhysics and followed the absorption/adsorption of water vapors in the SiO_2_ layer. The simulation coupled the “Chemical Species Transport of Diluted Species” with “Solid Mechanics” modules. It is a stationary analysis and computed the mechanical stress that appears due to diffusion [43]. The simulation neglected the piezoelectric and the thermal effects.

Due to the periodicity of the sensor device, the geometry used in the simulations is reduced by one wavelength (λ = 680 nm). All geometrical parameters are shown in Figure 1.

Figure 2 presents the result of the coupled simulation, showing the displacement due to humidity. The displacement appears on top of the SiO_2_ layer and increases by an order of magnitude in the case of a relative humidity of 90% compared to a relative humidity of 10%.

Figure 3 shows the average values of the von Mises stress versus the variation in the relative humidity, between 10% and 90%. These average values (calculated in the SiO_2_ layer) have a linear behavior, increasing when the humidity value increases, subsequently changing the values of the resonance frequency.

## 3. Experimental Details

We used a single-port SAW-type resonator structure, fabricated on a GaN/SiC layered structure. The piezoelectric layer we used was 1 µm thin GaN film deposited by MOCVD techniques on a SiC substrate. The GaN/SiC wafers were obtained on a commercial basis from NTT-AT, Tokyo Japan. The fabrication process involved the use of 3 photolithographic masks and an EBL (Electron Beam Lithography) process, as follows:-first, the CPW (Coplanar Wave Guide) line necessary for connecting the active area, formed by the IDT area, is defined; the metallization used in this case was Ti/Au, with a total thickness of ~100 nm (10 nm Ti/90 nm Au);-the next step consisted of the fabrication of the IDT area; for this, an EBL process was used, with the dimensions of the digits and the distance between them being 170 nm (λ = 680 nm); for the manufacture of IDTs, the metallization was TiAu, with a thickness of ~50 nm (5/45 Ti/Au);-next, an overlayer with a thickness of ~250 nm (20 nm Ti/230 nm Au) is deposited over the CPW line, which had the role to ensure the connection between the IDTs and the CPW line, but also to reduce losses;-the last step defined the functionalization layer above the IDTs; the deposition of the SiO_2_ layer is performed by RF sputtering and had a thickness of ~50 nm.

To observe the effect of the number of pairs of electrodes of the IDT on the quality factor Q, the structures were fabricated in two versions, which differed in the number of pairs: 75 and 150 pairs; in both cases, the wavelength used was 680 nm. Photos of the manufactured resonators are presented in Figure 4.

## 4. SAW Sensor Characterization

The characterization of the fabricated sensors was carried out in two stages: in the first stage, they were measured on-wafer in order to evaluate the resonance frequencies and the effect of the deposited functionalization layer on the resonance frequency; in the second stage, the humidity sensors were measured to determine the sensitivity. In this stage, the objective was to characterize the fabricated SAW sensor structures for the determination of the resonance frequencies and the effect of the number of pairs composing the SAWs has on the quality factor.

### 4.1. Room Temperature S Parameter Measurements of the Manufactured SAW Structures

The characterization was performed on-wafer, at room temperature, and before and after the deposition of the functionalization layer for humidity measurement. On-wafer measurements were performed by a Vector Network Analyzer 37397D (Anritsu, Morgan Hill, CA USA) and a pair of PM5 probes for on-wafer measurements from Suss MicroTec, Garching Germany.

The measurements made at room temperature showed a significant increase in the quality factor with the number of pairs. The best results obtained, presented in Figure 5, show us a Q factor of 831 for an SAW structure with 75 electrode pairs and 1445 for 150 pairs. Measurements of the resonance frequencies, for more than 10 resonators of each type, show that the resonance was between 6.66 GHz and 6.69 GHz for the Rayleigh mode, meaning a deviation lower than 0.5%, while for the Sezawa mode, the measurements show resonance frequencies between 7.99 GHz and 8.05 GHz, with a deviation of ~0.75%.

After functionalization layer deposition (SiO_2_, by RF sputtering, ~50 nm), the structure was measured again, in the same conditions. Figure 6 shows measurements of the same structure, before and after functionalization layer deposition; there was a slight decrease in the Rayleigh resonance frequency (from 6.66 GHz to 6.62 GHz) and in the amplitude (from ~−11.15 dB to ~−9.15 dB), but the Q factor in this case increased by ~250 (from 1445 to 1697). For the Sezawa mode, the resonance frequency decreased from 7.99 GHz to 7.83 GHz, and the amplitude remained at about the same value; the Q factor also increased from 223 to 290.

As wireless data transmission is envisaged, the electromechanical coupling coefficient, $k_{eff}^{2}$ which can be calculated using(5)$$k_{eff}^{2}=\frac{\pi^{2}}{4}\cdot \frac{f_{a}-f_{r}}{f_{a}}$$
and the fractional bandwidth, *FBW*, defined as(6)$$FBW=\frac{f_{a}-f_{r}}{f_{a}}\cdot 100(\%)$$
were also important and were analyzed. In (5) and (6), *f_r_* and *f_a_* represent the resonance frequency and the antiresonance frequency, respectively.

The values for *f_r_* and *f_a_* were extracted from the data of the S parameter measurements for the analyzed structure for the Rayleigh and Sezawa modes. The values for Q, $k_{eff}^{2}$, and *FBW* for the SAW structure used as humidity sensors are summarized in Table 1.

### 4.2. Characterization of the Humidity SAW Sensors

As the SAW device has a significant dependence on the resonance frequency of the temperature variation, first, we measured the temperature dependence of the resonance frequency of the SAW structure, containing the SiO_2_ functionalizing layer (Figure 7). SAW structures as temperature sensors were intensively analyzed by our team [44,45]. In particular, the structure used as a humidity sensor had an additional SiO_2_ layer, so its behavior vs. temperature variation was measured. In Figure 7, the resonance frequency for the Rayleigh mode (left) and Sezawa mode (right) is presented. When RH = 40%, there was a relatively high variation in the frequency vs. temperature. That temperature had to be maintained constant with high precision.

For the humidity sensors, characterization was performed using a VS 55240/LS device (TIRA, Schalkau Germany), having a climatic chamber of 64 L and allowing a temperature range of 30 °C … +150 °C and a humidity range of +10% … +95% RH (±3% … ± 5% RH).

For humidity measurements, the structures were cut into chips, mounted on specially manufactured carriers, and connected to the VNA; measurements were performed in the humidity range of 20% ÷ 90%, using a step to increase and decrease it by 1%/min, to let the structure accommodate the change. Measurements started from a humidity of ~35% (humidity of the ambient), was increased up to ~90%, and decreased back to 20%; the step used for the measurements was ~5%; measurements were performed at 30 °C. Figure 8 shows the results obtained for the adsorption (in red) and desorption (in blue) of the humidity for the Rayleigh and Sezawa resonance frequencies.

Measurements show very good results in terms of sensitivity: for the Rayleigh mode, the sensitivity measured was 6.17 kHz/%, while for humidity desorption, it was 3.79 kHz/%. As was expected, the sensitivity for the Sezawa mode was higher: 17.2 kHz/% for adsorption and 8.88 kHz/% for desorption.

It must be noticed that the differences between the adsorption and desorption sensitivity values can be explained due to the fact that some vapors are adsorbed and some are absorbed by the functionalization layer, SiO_2_.

A comparison with recently reported values for sensitivities is presented in Table 2. The results presented in Table 2 prove that the values for sensitivity obtained in this work represent state-of-the-art values. The use of GHz operating SAW devices based on “slow on fast” III-Nitride-based structures (where the Sezawa mode also appears) as humidity sensors permits the obtaining of excellent values for sensitivity.

We repeated the measurements of the humidity sensor after one month, checking that the temperature was 30 °C for all points, and we obtained the same results, with an error of less than 3%.

## 5. Conclusions

In this paper, we presented the fabrication and characterization of surface acoustic wave sensors with a resonant frequency higher than 6 GHz and with a high quality factor. The fabricated structures were first characterized at room temperature, with and without the functionalization layer, and revealed resonant frequencies of ~6.67 GHz for the Rayleigh mode and ~8 GHz for the Sezawa mode. The structures were fabricated in two versions, with 75 and 150 pairs of electrodes, and we observed a significant improvement in the quality factor: ~250 in the case of the Rayleigh mode and ~70 for the Sezawa mode; this means an improvement of approximately 17% in the first case and over 30% in the second. 

Humidity measurements of sensors manufactured using SiO_2_ as the functionalization layer have shown state-of-the-art values for sensitivity. The use of GHz operating SAW devices based on GaN/SiC structures (where the Sezawa mode also appears) as humidity sensors permitted the obtaining of very high values for sensitivity.

## Figures and Tables

**Figure 1 micromachines-16-00150-f001:**
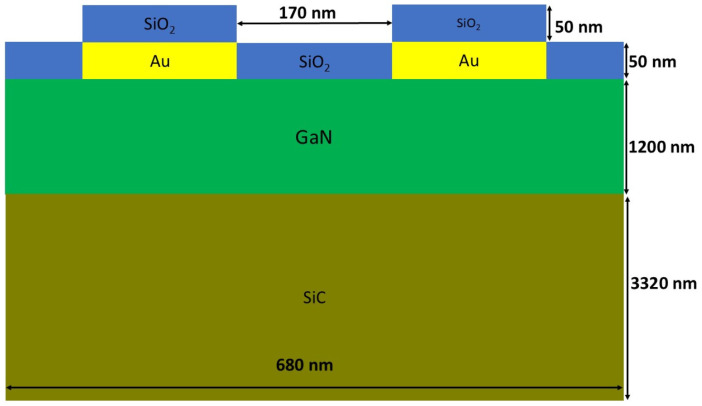
Geometrical parameters and materials used in simulation.

**Figure 2 micromachines-16-00150-f002:**
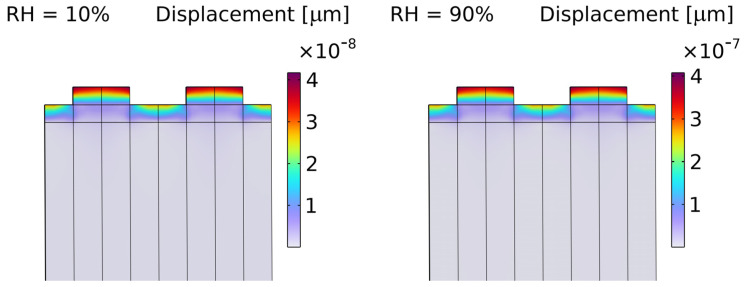
The displacement induced by humidity when relative humidity (RH) is 10% (**left**) and 90% (**right**).

**Figure 3 micromachines-16-00150-f003:**
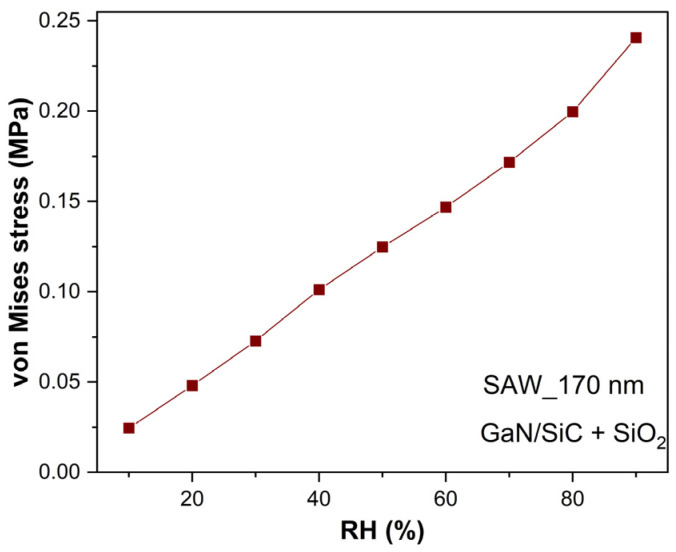
Equivalent stress function of relative humidity.

**Figure 4 micromachines-16-00150-f004:**
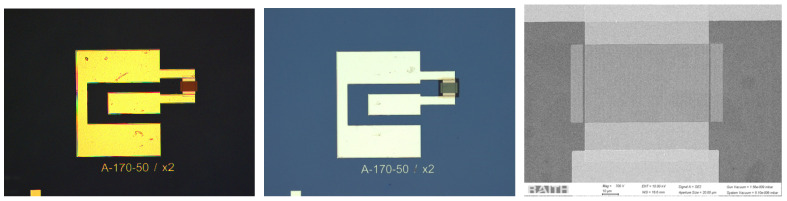
Photos of the the SAW resonator (170 nm electrode width; 150 electrode pairs) before (**left**) and after SiO2 layer deposition (**middle**); SEM photo of the IDTs area (**right**).

**Figure 5 micromachines-16-00150-f005:**
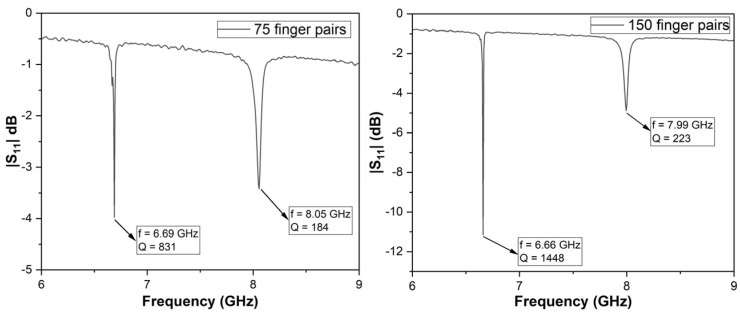
Measured S_11_ parameters of the SAW resonators (75 electrode pairs—**left**; 150 electrode pairs—**right**) at room temperature.

**Figure 6 micromachines-16-00150-f006:**
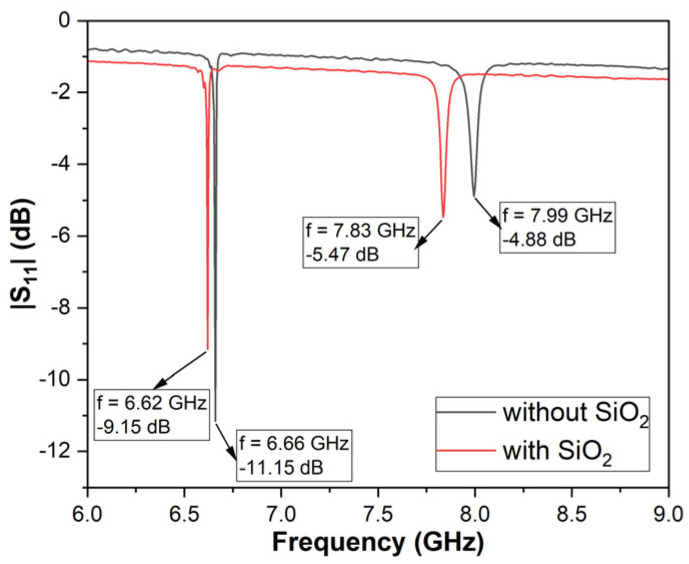
S_11_ parameter of the SAW resonator vs. frequency after SiO_2_ layer deposition.

**Figure 7 micromachines-16-00150-f007:**
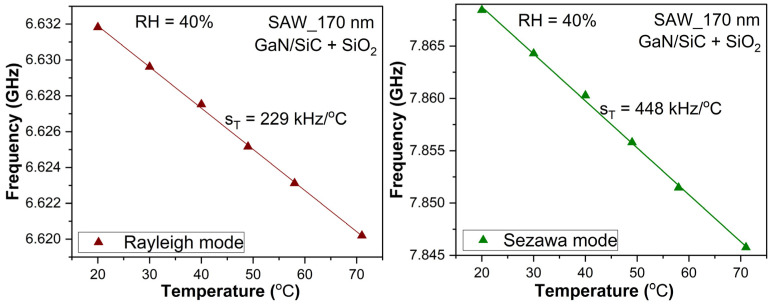
Variation in resonance frequency for SAW with the SiO_2_ functionalization layer vs. temperature for the Rayleigh mode (**left**) and for the Sezawa mode (**right**); the temperature sensitivity is defined as s_T_ = df_res_/dT.

**Figure 8 micromachines-16-00150-f008:**
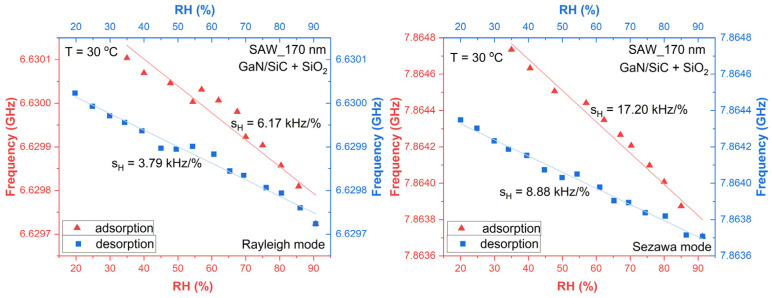
Variation in the resonance frequency for adsorption and desorption of the humidity for Rayleigh (**left**) and Sezawa (**right**) resonance frequencies.

**Table 1 micromachines-16-00150-t001:** Experimental values for Q, $k_{eff}^{2}$, and *FBW* for the SAW structure used as humidity sensors.

Parameter (Unit)	Symbol	Rayleigh	Sezawa
Quality factor	Q	1697	290
Coupling coefficient (%)	$k_{eff}^{2}$	0.37	1.1
Fractional bandwidth (%)	*FBW*	0.15	0.44

**Table 2 micromachines-16-00150-t002:** Sensitivities of SAW humidity sensors on different substrates based on various sensing materials.

Sensing Material	Substrate	RH (%) Range	Sensitivity (kHz/%RH) *	References
Sputtering SiO_2_ film	GaN/SiC	35 … 90	6.17 (Rayleigh) 17.20 (Sezawa)	present work
GO/TiO2	Quartz	10 … 90	13.945	[34]
Sol–gel SiO_2_ film	Quartz	30 … 93	7.46	[35]
ZnO nanorods	AlN	30 … 90	3.77	[46]
Fluorinated polyimide PI	AlN	10 … 90	4.15	[47]
Sputtering SiO_2_ film	Quartz	10 … 80	1.14	[48]
ZnO nanoparticles	Quartz	20 … 95	0.237 (Rayleigh) 0.388 (Sezawa)	[49]

(*) Desorption values are not considered in Table 2.

## Data Availability

The original contributions presented in this study are included in the article. Further inquiries can be directed to the corresponding author.

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
