# Peer review of "High Q GaN/SiC-Based SAW Resonators for Humidity Sensor Applications"

_micromachines, 2025, doi:10.3390/mi16020150_

Round 1
Reviewer 1 Report
Comments and Suggestions for Authors
The paper is devoted to development of super high frequency SAW humidity sensors based on one port resonator with sensitive layer SiO2. The paper is interesting and in Scope of Micromachines. I think authors should pay attention to the next points:
1. Introduction is very short. As I know there are exist many fresh paper devoted to different kind of acoustoelectronic humidity sensors.
2. Authors deposited SiO2 sensitive layer on IDT. What about repeatability of measurements? Do the properties of the electrode structure change due to contact with a wet film?
3. What is the physical mechanism of SAW sensitivity to humidity in your sensor?
4. It is necessary to compare obtained sensitivity with sensitivity other SAW humidity sensors.
Author Response
Reviewer 1
The paper is devoted to development of super high frequency SAW humidity sensors based on one port resonator with sensitive layer SiO2. The paper is interesting and in Scope of Micromachines. I think authors should pay attention to the next points:
- Introduction is very short. As I know there are exist many fresh papers devoted to different kind of acoustoelectronic humidity sensors.
The introduction has been significantly extended (all paragraphs in red), addressing more details related to the state of the art and the development of the paper, as well as answers to some reviewer questions and remarques.
- Authors deposited SiO2 sensitive layer on IDT. What about repeatability of measurements? Do the properties of the electrode structure change due to contact with a wet film?
The role of the SiO2 layer is, first of all, to adsorb humidity (the water molecules) from the environment and secondly to insulate the IDT’s structure of metal electrodes, to avoid their short-circuiting.
This was detailed in the revised version of the paper in Chapter 1 Introduction, last paragraph, in red
We haven’t observed any changes in contacts’ behavior due to humidity.
We have repeated the measurements of the humidity sensor, after about one month and we have obtained the same results within an error of lower than 3%.
This was evidenced in last phrase of Chapter 4.
- What is the physical mechanism of SAW sensitivity to humidity in your sensor?
The main physical mechanism of SAW interaction with the humidity is the mass loading mechanism, having as effect the change of the phase velocity. The fractional frequency changes are equal with the fractional velocity changes and the equation for the mass-induced change in SAW propagation velocity is, ∆f/f0 =∆v/v0 = -cmf0 ∆m/A, where, is mass sensitivity factor that is a material constant specific to the piezoelectric layer and ∆m/A is the mass change per surface unit of the adsorbed/absorbed gas.
The sensitivity calculated for a SAW device as gas sensor is S = d∆f/d(∆m/A )= - cmf0 2
The sensitivity of mass loading-based devices is directly proportional to the square of its operating frequency for gas and humidity detection SAW sensors. For humidity sensors, the sensing performance is quantified as absolute sensitivity (Sa = Δf/ΔRH) and relative sensitivity (Sr = Sa/f0), (ΔRH is change in relative humidity).
This explanation has been detailed in the revised version of the paper in the Introduction chapter, 1, paragraphs 8 and 9.
- It is necessary to compare obtained sensitivity with sensitivity other SAW humidity sensors.
A comparison of the parameters of our sensors with other humidity SAW sensors was included in the paper and is presented in Table 1, at the end of chapter 4, SAW sensors characterization
Other minor changes:
The abstract was reformulated for a better English language.
In the conclusions chapter a sentence was added (in red) and another was removed.

Reviewer 2 Report
Comments and Suggestions for Authors
In this manuscript, a high Q SiO2/GaN/SiC SAW resonators were fabricated and the sensitivities of its resonance frequency to humidity for Rayleigh-type SAW and Sezawa wave were experimentally investigated.
However, at the present stage, this manuscript is judged should not be published in Micromachines because this manuscript has several problems as follows:
(1) The motivation for using GaN thin film and SiC substrate is not clearly stated.
(2) There is no description of the consideration for the simulation results of Fig. 2
(3) There is no comparison or discussion between the simulation results in Section 2 and the experimental results in Section 3.
(4) Although the Q value increases or decreases depending on the fractional bandwidth of the resonance property, the fractional bandwidth has not been investigated.
(5) Figure 6, which is a photograph of a ready-made product, should be omitted.
(6) The frequency change with temperature, which is important for humidity sensors, has not been considered.
Author Response
Reviewer 2
In this manuscript, a high Q SiO2/GaN/SiC SAW resonators were fabricated and the sensitivities of its resonance frequency to humidity for Rayleigh-type SAW and Sezawa wave were experimentally investigated.
However, at the present stage, this manuscript is judged should not be published in Micromachines because this manuscript has several problems as follows:
- The motivation for using GaN thin film and SiC substrate is not clearly stated.
Most of works on SAW based sensors was focused on SAW devices manufactured on classical piezoelectric substrates such as quartz, langasite or lithium niobate. These materials have excellent piezoelectric properties but the quality of their surface make very difficult the development on their surface of advanced nanolithographic processes.This limits the resonance frequencies of most SAW devices manufactured on this material to 2.5 GHz. An increase of the SAW resonance frequency is important many applications eg. filters for 5G and above communication applications and also in sensor applications as higher resonance frequency it enhances the sensitivity of the SAW sensor. Piezoelectric III-Nitride based layered structures represents an interesting alternative for high frequency SAW devices.
Gallium Nitride (GaN) a III-nitride type semiconductor is also a piezoelectric material, with excellent piezoelectric properties. Layered structures GaN/Si, GaN/SiC and GaN/Sapphire are fully compatible with nanolithography and micromachining process that can be introduced in the fabrication protocol of the acoustic devices. In the last years, these piezoelectric substrates became attractive for high sensitivity temperature and pressure sensors based on acoustic devices. For the experiments we will present in this work we have chosen to use GaN/SiC as it has two advantages compared with GaN/Si structures: (i) the higher quality of the GaN layer deposited on SiC, compared with silicon due to much less problems regarding the lattice mismatch (the lattice constant for GaN is 3.1 Å, 3.07 Å for SiC and 5.4 Å for Si) during the MOCVD growing process. (ii) the potential possibility to use the sensor at high temperatures.
This motivation is detailed in the revised version of the paper in chapter 1 Introduction paragraph 7.
- There is no description of the consideration for the simulation results of Fig. 2.
Explanation regarding the simulation method was added in the Chapter 2, paragraph 3
In the revised version, Figure 2 was replaced, where it is presented the distribution of the displacement for two humidity values, and a new figure was added (Figure 3), where the average values for von Mises stress was calculated.
- There is no comparison or discussion between the simulation results in Section 2 and the experimental results in Section 3.
At the current stage of the simulation procedure for the SAW humidity sensors we can provide the results regarding the displacement and the stress induced by the variation of the humidity. In order to obtain a real comparison between the experimental results and simulations in terms of resonance frequency variation with humidity, we need to couple the diffusion effect of humidity with the piezoelectric effect; this would be a very complex procedure that require computational effort as well as the software capacity to couple the required modules.
(4) Although the Q value increases or decreases depending on the fractional bandwidth of the resonance property, the fractional bandwidth has not been investigated.
The fractional bandwidth, FBW, according to our knowledge, is important for SAW devices used as filters. In our case we have analyzed SAW sensors so we have not analyzed this parameter.
(5) Figure 6, which is a photograph of a ready-made product, should be omitted.
Figure 6 has been deleted.
(6) The frequency change with temperature, which is important for humidity sensors, has not been considered.
In the revised version, we have added, at the beginning of sub-Chapter 4.2, an analysis of the temperature dependence of resonance the frequency of the SAW structure (containing the SiO2 functionalizing layer) (Fig 7 in the revised version) for the Rayleigh as well as for the Sezawa mode at a constant value of the humidity.
The resonance frequency changes are significant when temperature changes SAW structures as temperature sensors have been intensively analyzed by our team {xx]. In particular the structure used as humidity sensor has an additional SiO2 layer was now. The relatively high variation of the resonance frequency vs. temperature makes necessary to maintain the temperature during the humidity measurements constant with a high degree of precision This was also mentioned in chapter 4.2 in the revised version.
Other minor changes:
The abstract was reformulated for a better English language.
In the conclusions chapter a sentence was added (in red) and another was removed.
Round 2
Reviewer 1 Report
Comments and Suggestions for Authors
Thank you for paper correction.
Author Response
Thank you.
Reviewer 2 Report
Comments and Suggestions for Authors
Since wireless data transmission is expected as described in Introduction, it is necessary to indicate the electromechanical coupling factor, i.e., the fractional bandwidth (FBW), of the SAW resonator. Please describe the values of FBW of the Rayleigh-type SAW and Sezawa wave of the fabricated samples obtained from the following formula.
FBW=(fa-fr)/fa*100 [%]
Here, fa and fr are the antiresonance and resonance frequencies.
Author Response
[Comment1] Since wireless data transmission is expected as described in Introduction, it is necessary to indicate the electromechanical coupling factor, i.e., the fractional bandwidth (FBW), of the SAW resonator. Please describe the values of FBW of the Rayleigh-type SAW and Sezawa wave of the fabricated samples obtained from the following formula.
FBW=(fa-fr)/fa*100 [%]
Here, fa and fr are the antiresonance and resonance frequencies.
[Response1] Thank you for suggestion. Values for electromechanical coupling coefficient ,fractional band width have been determined from experimental data analyzed and summarized together with the Q factor in new Table 1. This analysis is added at the end of subsection 4.1 last paragraphs IN RED